# Transcriptome Studies Reveal the *N*^6^-Methyladenosine Differences in Testis of Yaks at Juvenile and Sexual Maturity Stages

**DOI:** 10.3390/ani13182815

**Published:** 2023-09-05

**Authors:** Shaoke Guo, Jie Pei, Xingdong Wang, Mengli Cao, Lin Xiong, Yandong Kang, Ziqiang Ding, Yongfu La, Min Chu, Pengjia Bao, Xian Guo

**Affiliations:** 1Key Laboratory of Yak Breeding Engineering of Gansu Province, Lanzhou Institute of Husbandry and Pharmaceutical Sciences, Chinese Academy of Agricultural Sciences, Lanzhou 730050, China; gsk1125@163.com (S.G.); peijie@caas.cn (J.P.); wxd17339929758@163.com (X.W.); caomengliaaa@163.com (M.C.); xionglin@caas.cn (L.X.); kangyandong0901@163.com (Y.K.); dingziqiang1997@163.com (Z.D.); layongfu@caas.cn (Y.L.); chumin@caas.cn (M.C.); baopengjia@caas.cn (P.B.); 2Key Laboratory of Animal Genetics and Breeding on Tibetan Plateau, Ministry of Agriculture and Rural Affairs, Lanzhou 730050, China

**Keywords:** spermatogenesis, mRNA, *N*^6^-methyladenosine, reproduction

## Abstract

**Simple Summary:**

The methylation level of testicular tissue in young yaks and adult yaks was determined, and the *N*^6^-methyladenosine (m^6^A) level was significantly increased after sexual maturity. Functional enrichment analysis of differentially methylated mRNAs screened between the two groups after sequencing showed that they mainly regulated cell development, spermatogenesis, and testicular endocrine function. Functional analysis of the differentially expressed genes screened between the two groups showed that they were involved in the biological processes of mitosis, meiosis, and flagellated sperm motility. Finally, several genes related to spermatogenesis were screened for further verification.

**Abstract:**

Studying the mechanism of spermatogenesis is key to exploring the reproductive characteristics of male yaks. Although *N*^6^-methyladenosine (m^6^A) RNA modification has been reported to regulate spermatogenesis and reproductive function in mammals, the molecular mechanism of m^6^A in yak testis development and spermatogenesis remains largely unknown. Therefore, we collected testicular tissue from juvenile and adult yaks and found that the m^6^A level significantly increased after sexual maturity in yaks. In MeRIP-seq, 1702 hypermethylated peaks and 724 hypomethylated peaks were identified. The hypermethylated differentially methylated RNAs (DMRs) (CIB2, AK1, FOXJ2, PKDREJ, SLC9A3, and TOPAZ1) mainly regulated spermatogenesis. Functional enrichment analysis showed that DMRs were significantly enriched in the adherens junction, gap junction, and Wnt, PI3K, and mTOR signaling pathways, regulating cell development, spermatogenesis, and testicular endocrine function. The functional analysis of differentially expressed genes showed that they were involved in the biological processes of mitosis, meiosis, and flagellated sperm motility during the sexual maturity of yak testis. We also screened the key regulatory factors of testis development and spermatogenesis by combined analysis, which included *BRCA1*, *CREBBP*, *STAT3*, and *SMAD4*. This study indexed the m^6^A characteristics of yak testicles at different developmental stages, providing basic data for further research of m^6^A modification regulating yak testicular development.

## 1. Introduction

Yaks (*Bos grunniens*) are an animal genetic resource known as “the boat of the plateau” and “the omnipotent livestock” in the Tibetan areas of China. They live in high-altitude areas above 3000 m and can tolerate cold and hypoxic conditions [1,2]. Yaks play a crucial role in the life, production, socio-economic development, and agricultural biodiversity conservation in plateau areas [3]. However, seasonal breeding patterns, low conception rates, and long calving intervals seriously restrict yak reproductive performance. Therefore, the study of yak reproductive performance is helpful in improving the economic development and people‘s quality of life in the Qinghai–Tibet Plateau.

The testis is the primary sexual organ of male mammals, responsible for androgen synthesis and secretion and spermatogenesis. The coordinated process of spermatogenesis involves many molecular and cellular events [4,5,6]. Establishing a relation between spermatogenesis and gene expression profiles can explore testis development processes to improve testicular function for high-quality semen production, thereby increasing the chances of obtaining healthy offspring. Multiple programs synergistically regulate specific gene expression at different stages of spermatogenesis [7]. Therefore, the study of the transition process of the testis from the growth stage to the spermatogenesis stage is key to understanding the reproduction and development of male animals.

*N*^6^-methyladenosine (m^6^A) is the most abundant internal modification of mRNA and a key mediator of gene expression control from yeast to animals and plants [8,9,10]. m^6^A modification can affect transcription, splicing, localization, translation, stability, and post-transcriptional regulation of gene expression at the RNA level [11]. RNA m^6^A regulators maintain dynamic m^6^A modification [12], where mRNA undergoes methylation modification under the action of writers such as methyltransferase-like protein 3 (METTL3), methyltransferase-like protein 14 (METTL14), and Wilms’ tumor 1-associated protein (WTAP), and is then demethylated by erasers such as fat mass and obesity-associated protein (FTO) and alkB homolog 5 (ALKBH5). Readers, including YTH domain family proteins 1–3 (YTHDF1–3) and YTH domain-containing proteins 1–2 (YTHDC1–2), can recognize the bases of m^6^A methylation, participate in downstream translation and mRNA degradation, and accelerate the speed of mRNA nucleation [13]. m^6^A plays an important regulatory role in cell fate transition and animal development [14].

In particular, m^6^A modification is involved in spermatogonial proliferation, spermatogenesis, and reproductive function in mammalian reproduction [15,16,17]. Studies have reported that m^6^A regulators are expressed in both spermatogenic cells and somatic cells [18]. ALKBH5 in the testis regulates RNA metabolism, and it is mainly involved in the regulation of alternative splicing of mRNA in spermatogenic cells [19,20]. ALKBH5 and METTL14 are expressed in spermatogenic cells and interstitial cells and affect the synthesis and secretion of testosterone [21]. Lin et al. [22] showed that inhibition of METTL3/ METTL14 activity, which can reduce m^6^A and disrupt the proliferation of spermatogonial stem cells, hindering spermatogenesis. YTHDC2 is essential for the gene expression program of the male germline from mitosis to meiosis [23,24], and YTHDC1 is essential for maintaining spermatogonial development in mice [25]. Liu et al. [26] performed m^6^A-seq and RNA-seq on porcine spermatogonia, spermatocytes, and round sperm cells and analyzed the mechanism of m^6^A in spermatogenesis. Wang et al. [27] compared the m^6^A modification sites of yak and cattle–yak testis and discussed the effect of m^6^A on spermatogenesis arrest of cattle–yak. Liu et al. [28] analyzed the expression spectra of m^6^A in bovine testis at different developmental stages by MeRIP-seq analysis and identified several key regulatory genes related to the regulation of mammalian testis development and spermatogenesis.

Although the mechanism by which m^6^A regulates bovine spermatogenesis has been reported, the impact of m^6^A in yak testis development and spermatogenesis remains largely elusive. To further investigate the regulatory role of m^6^A on spermatogenesis in yak testis, we collected 6-month-old and 30-month-old yak testicular tissue samples for MeRIP-seq. This allowed us to obtain the m^6^A spectrum of the transcriptome at different developmental stages of yak testis. Our study provides a theoretical basis for the role of m^6^A during spermatogenesis in testis development after birth, as well as preliminary data for the research of the yak reproduction mechanism essential to yak breeding.

## 2. Materials and Methods

### 2.1. Ethics Statement

All animal procedures were performed according to the China Council on Animal Care guidelines and the Ministry of Agriculture of the People’s Republic of China. The yak handling procedures were approved by the Animal Care and Use Committee of the Lanzhou Institute of Husbandry and Pharmaceutical Sciences Chinese Academy of Agricultural Sciences (Permit No: SYXK-2014-0002).

### 2.2. Sample Collection

Six healthy male yaks from Xiahe County, Gannan Tibetan Autonomous Prefecture (34°51′ N, 102°26′ E), were selected. According to the degree of sexual maturity, three 6-month-old yaks (Y6M) before sexual maturity were selected for the first group, and three 30-month-old yaks (Y30M) were selected for the second group; the samples were collected from yaks during the estrus period in August. After local disinfection of the testes, they were castrated by veterinarians to obtain testicular tissue. The collected testicular tissue was washed clean with normal saline, and a scalpel was used to draw along the longitudinal axis of the testis. The testicular parenchyma vertical with the longitudinal axis was collected, and the cut tissue was trimmed into a tissue block with a volume of about 1 cm^3^, which was fixed in 10% neutral formaldehyde for producing tissue sections, and the consistency of the collection site was maintained. The remaining tissue was cut into three parts and put into a clean cryopreservation tube for liquid nitrogen cryopreservation.

### 2.3. Library Sequencing and Bioinformatics Analysis

The steps of RNA extraction and quality control, library construction, sequencing, and bioinformatics analysis refer to the methods reported by Guo et al. [29]. The levels of m^6^A and m^6^A-related enzymes in the total RNA of testicular tissue were determined (Appendix A). Qualified RNA was selected for reverse transcription into cDNA, and oligo-dT magnetic beads were used for two rounds of purification, and mRNA was specifically captured with polyadenylate for library construction and sequencing. Filter out low-quality and N-base or low-quality readings and obtain high-quality clean readings. The clean readings were compared with the yak reference genome (LU_Bosgru_v3.0), and only aligned readings were retained for subsequent analysis. MeTDiff [30] software was used to perform peak detection with input samples as control. The detected peak was annotated with ChIPseeker [31] software. Differential peak detection analysis was performed using MeTDiff software (*p* ≤ 0.05; |FC| ≥ 1.5). In the transcriptome sequencing analysis, the fragments per kilobase of transcript per million mapped reads (FPKM) value of the gene was used to represent the expression level of the gene. The screening condition for differentially expressed genes between samples was (*p* ≤ 0.05; |FC| ≥ 1.5). The condition for screening significantly enriched items in Gene Ontology (GO) and Kyoto Encyclopedia of Genes and Genomes (KEGG) analysis was *p* < 0.05. Detailed experimental steps and data analysis process are shown in Appendix A.

### 2.4. Verification of Differential Gene Expression

According to the gene expression results, *BRCA1*, *CCNE2*, *SOX5*, *PDE4A*, *INHBA*, *ERBB4*, *IGFBP4*, and *SMAD4* were selected to verify the sequencing results by qRT-PCR. Data were expressed as mean ± standard error (*n* = 3). The glyceraldehyde-3-phosphate dehydrogenase gene was used as an internal reference gene, and the 2^−ΔΔCT^ calculation method was used to standardize mRNA expression [32]. All statistical analyses were performed using SAS 9.4 (2013) statistical software, and *p* ≤ 0.05 was considered statistically significant. The primer synthesis of the target gene was performed using the NCBI website (Appendix A).

## 3. Results

### 3.1. Morphological Changes in Yak Testis before and after Sexual Maturity

Morphological observations of the tissue sections of yak testis at the juvenile stage (6-month-old, Y6M) and sexual maturity stage (30-month-old, Y30M) are shown in Figure 1a. The testicular volume of juvenile yaks was small, the testicular seminiferous tubules were sparsely arranged, only spermatogonia and Sertoli cells were visible in the seminiferous tubules, and single or a group of Leydig cells were visible in the stroma. The seminiferous tubules had a fine diameter, with an average diameter of 76.8 μm, and no lumen was observed. The tubular interface membrane was clear, and only 1–2 layers of spermatogonia were regularly arranged in the basement membrane of the spermatogenic epithelium. The testicular volume of sexually mature yaks significantly increased, the seminiferous tubules were densely arranged, and the interstitial cells were densely distributed in groups, and the number significantly increased. The diameter of seminiferous tubules significantly increased, with an average diameter of 193.1 μm. The number of spermatogenic cell layers increased, and the cells were orderly arranged in 4–6 layers from the basement membrane to the lumen surface. After sexual maturity, the number of spermatogonia and spermatocytes in the testicles of yaks increases, and more round sperm cells and spermatids are observed near the lumen.

### 3.2. Yak Testis m^6^A Level before and after Sexual Maturity

To clarify whether m^6^A plays a regulatory role in yak testis development, the m^6^A levels of testicular tissue before and after sexual maturity were first compared. The m^6^A level of yak testicular tissue after sexual maturity was significantly (*p* < 0.05) higher than that at the juvenile stage (Figure 1b). The expression levels of 12 m^6^A-related enzymes were detected by quantitative real-time PCR (qRT-PCR). The expression of *FTO* and *ALKBH5* was significantly (*p* < 0.01) higher in sexually mature yak testis than in juvenile yak testis (Figure 1d), and the expression of all recognition proteins detected was also significantly (*p* < 0.01) increased (Figure 1e). However, the expression of *METTL3*, *WTAP*, and *VIRMA* showed no significant (*p* > 0.05) difference in yak testis before and after sexual maturity (Figure 1c). Thus, the demethylation process of m^6^A modification plays an important regulatory role in yak testis development.

### 3.3. Raw Data Quality Control and Characteristics of m^6^A in Yak Testis

To further understand m^6^A methylation during testis development, MeRIP-seq was performed on juvenile and sexually mature yak testicular tissue samples, with three replicates in each group. A total of 91.61 Gb of clean data were obtained from MeRIP-seq and input sequencing of six testicular samples, and the average effective data volume of each sample was 7.63 Gb (Appendix A). By using HISAT2 [33], the clean reads were aligned with the yak reference genome (LU_Bosgru_v3.0), resulting in uniquely mapped reads that were 88.22–93.39% of total clean reads (Figure 2a). Reads with multiple alignments were excluded, and only the aligned reads were retained for subsequent analysis. The MeTDiff [30] software was used to perform peak calling analysis. A total of 31,100 m^6^A peaks were detected in this sequencing, and 15,393 peaks were identified in juvenile yak testicular tissue. The average length of peaks was 6444.22 bp (Figure 2b), accounting for 3.50% of the yak genome (Appendix A). A total of 15,707 peaks were identified in sexually mature yak testicular tissue, with an average length of 5766.50 bp, accounting for 3.20% of the yak genome (Appendix A).

Using ChIPseeker [31] software to annotate the peaks detected in the samples, we found that more than half of the peaks were located in the exon regions, and the peaks located in 5′and 3′UTRs accounted for 9.18% and 22.25%, respectively (Figure 2c). The m^6^A distribution pattern displayed that the distribution of m^6^A peaks on yak chromosomes was different, but the overall distribution density trend was similar, and m^6^A peaks were mostly concentrated on chromosomes 8, 19, and 20 (Figure 2d). Statistical analysis of the enrichment position of the m^6^A peak showed that its abundance was the highest near 3′UTR, followed by 5′UTR and the coding DNA sequence region (Figure 2e). Two mRNAs, RBM11 and EURL, with a methylated peak at 3′UTR and 5′UTR, respectively, were randomly selected for visual observation (Figure 2f).

The m^6^A modified sequence has certain motif characteristics, which may be related to major regulatory regions and biological functions. MEME-ChIP [34] was used to analyze and identify the motif sequence in the peak sequence. In yak testicular tissue, m^6^A mainly occurred in the core motif sequence “AAACH” (Figure 2g), which shows the conservative pattern of the “RRACH” motif, indicating that the peak identified was highly reliable.

### 3.4. Analysis of Differentially Methylated Peaks (DMPs) of Yak Testis before and after Sexual Maturity

To explore the potential role of the m^6^A modification in yak testis, differentially methylated peaks (DMPs) were identified in Y6M and Y30M samples (Appendix A). A total of 2426 significant DMPs were identified, including 1702 hypermethylated peaks and 724 hypomethylated peaks (Figure 3a). The overall distribution of DMPs is shown in Figure 3b. Functional enrichment analysis was performed on hypermethylated and hypomethylated differentially methylated RNAs (DMRs) [35,36]. GO analysis showed that DMRs were mainly annotated in the nucleus, nucleoplasm, and histone methyltransferase complex in the cellular component category (Figure 3c). DMRs were significantly enriched in DNA binding, DNA-binding transcription factor activity, RNA polymerase II-specific, and microtubule binding in the molecular function category. In the biological process category, DMRs were significantly enriched in the regulation of gene expression and negative regulation of the canonical Wnt signaling pathway. KEGG analysis showed that DMRs were enriched in 53 pathways, which were significantly correlated with Wnt signaling, the adherens junction, MAPK signaling, insulin resistance, PI3K–Akt signaling, and mTOR signaling pathways (Figure 3d). Enrichment analysis revealed that DMRs were involved in major male reproductive processes, including the development of testicular seminiferous tubules, spermatogonial differentiation, and meiosis. In addition, *CIB2*, *AK1*, *FOXJ2*, *PKDREJ*, *SLC9A3*, and *TOPAZ1* genes in the hypermethylated DMRs were specifically expressed in spermatocytes and round sperm cells, regulating spermatogenesis [37,38,39,40,41,42,43,44,45]. In hypomethylated DMRs, it was found that BCL6 is associated with cell apoptosis [46,47], EPHA2 regulates self-renewal of SSCs [48,49], TAF12 regulates the transcription program of specific genes in the testes [50,51].

### 3.5. Analysis of Differentially Expressed Genes (DEGs) in Yak Testis before and after Sexual Maturity

The box diagram of the FPKM value of each sample gene is shown below (Figure 4a). The overall gene expression level between the same group of samples was similar. The highest expression level of genes was slightly higher in the testis of juvenile yaks than that of sexually mature yaks. The density distribution of FPKM values of each sample gene showed that the number of genes with the FPKM value of 10–30 was the largest, and most genes had a moderately high expression level (Figure 4b). A total of 10,298 DEGs were identified in the experimental group (*p* < 0.05; |FC| > 1.5) (Appendix A), of which 5661 genes were significantly up-regulated, and 4637 genes were significantly down-regulated (Figure 4c,d). The hierarchical clustering expression mode of DEGs is shown in Figure 4e. GO analysis in the molecular function category showed that DEGs were significantly enriched in ATP binding, structural constituents of ribosomes, and calcium ion binding (Figure 4f). In the cellular component category, DEGs were mainly concentrated in acrosomal vesicles, the cytoskeleton, and motile cilium. In the biological process category, DEGs were significantly related to spermatogenesis, cell differentiation, and cilium assembly. KEGG enrichment analysis of DEGs showed that 90 pathways were significantly enriched, including ribosomes, the extracellular matrix (ECM)–receptor interaction, cortisol synthesis and secretion, aldosterone synthesis and secretion, the cAMP signaling pathway, and other pathways (Figure 4g). Enrichment analysis showed that differential genes were involved in spermatogenesis, germ cell proliferation, differentiation, and metabolic activities.

### 3.6. Combined Analysis of m^6^A-Seq and RNA-Seq Data

To explore the potential relationship between m^6^A modification and gene expression, a combined analysis of m^6^A-seq and RNA-seq data was performed (Appendix A). A total of 1476 co-DEGs were identified (*p* < 0.05; |FC| > 1.5). Among the 1013 m^6^A hypermethylated peaks, 581 mRNAs were up-regulated, and 432 genes were down-regulated. Among the 463 m^6^A hypomethylated peaks, 170 genes were up-regulated, and 293 genes were down-regulated (Figure 5a). The 30 co-expressed genes related to testis development with significant changes in the transcriptional abundance of m^6^A and mRNA in the yak testis are listed in Table 1. A gene regulatory network map was constructed using the String database (Figure 5b). Breast cancer 1 (BRCA1), recombinant CREB-binding protein (CREBBP), signal transducer and activator of transcription 3 (STAT3), and recombinant mothers against decapentaplegic homolog 4 (SMAD4) were the most connected key node factors and were reported to play a regulatory role in spermatogonial differentiation, meiosis, and spermatogenesis [52,53,54,55]. To verify the results of RNA-seq, eight genes, *BRCA1*, *CCNE2*, *SOX5*, *PDE4A*, *INHBA*, *ERBB4*, *IGFBP4*, and *SMAD4* were randomly selected for qRT-PCR analysis. The qRT-PCR results showed that the expression levels of these eight genes had a similar trend to the expression levels of RNA-seq (Figure 5c,d).

## 4. Discussion

The silver fox is a seasonal estrous animal. Cheng et al. [56] reported that the testis mass, volume, and testosterone content of silver foxes increased gradually from September to January of the next year. The composition and number of spermatogenic cells in the cross-section of a single seminiferous tubule on the section gradually increased with time and showed seasonal changes [56]. The yak is also a seasonal estrous animal, usually experiencing estrus during summer. The production and reproduction of male yaks are affected by genetic, environmental, and management factors, including season, light, temperature, nutrition, and breeding techniques [57]. Due to the long-term domestication, climate, and forage resources, yaks only have sexual behavior in the estrus season, which greatly limits the reproductive activity of male yaks. The testicular volume and weight of male yaks are lower than those of ordinary cattle [58], resulting in lower sperm production. In addition, yak breeds appeared degradation phenomenon due to inbreeding, improper feeding and management, and insufficient grassland quality. The efficiency of artificial insemination of yaks in pastoral areas is extremely low, which also poses challenges to yak breeding and production [59]. The low efficiency of yak reproductive performance not only affects the improvement of yaks and the breeding of new varieties but also limits the large-scale promotion of yaks, which has a certain impact on the economy of herdsmen and affects the development of animal husbandry economy in the Qinghai–Tibet Plateau.

Studying the reproductive physiology of male yak is important to understand its reproductive potential. It is essential to improve yak fertility, enhance yak reproductive levels, and breed new varieties. The mechanism of postnatal testis development in mammals has been a research hotspot. However, studies on the developmental stages of yak testis from growth to spermatogenesis are few. Epigenetic regulation, including DNA methylation, histone modification, and non-coding RNA regulation, is closely related to spermatogenesis [60]. The regulatory relationship between RNA m^6^A modification and spermatogenesis has been found in the testes of different species [61,62,63], representing another level in the regulatory network of spermatogenesis. Therefore, we selected yak testicular tissue before and after sexual maturity for MeRIP-seq, analyzed the gene expression process regulated by m^6^A during testis development, and identified the transition-related genes during testis development.

Spermatogenesis is a process of precise regulation of male germ cells from proliferation and differentiation to sperm production, including the proliferation of spermatogonia, the growth and maturation of spermatocytes, the processing of sperm cells, and the formation of sperm [64]. Sexual maturity generally occurs in male yaks at 2–3 years of age [65]. On tissue structure observation, we found that there were only primitive spermatogonia and supporting cells but no spermatocytes in the testis of 6-month-old yaks. We observed primary spermatocytes, secondary spermatocytes, round spermatids, and mature sperm in the testis of 30-month-old yaks, indicating that male yaks had reached sexual maturity after 2 years old. In addition, various somatic cells, including Sertoli and Leydig cells, play an important role in spermatogenesis. Leydig cells synthesize and secrete more than 90% of androgens in the body [66]. Sertoli cells regulate the secretory function of Leydig cells by interstitial activator proteins and secrete various growth factors to support the proliferation and differentiation of spermatogonia [67]. Of course, such a complex spermatogenesis process is subject to strict and precise molecular regulation. Lin et al. [22] showed that m^6^A modification with overall dynamic changes occurred at different developmental stages of spermatogenesis, which is consistent with our results of a significantly increased overall m^6^A level in the testis at sexual maturity. In our results, the expression of *FTO*, *ALKBH5*, and *YTHDC2* genes was significantly up-regulated in the testis of sexually mature yaks. ALKBH5-mediated m^6^A demethylation is essential for the splicing of m^6^A-modified long 3′ UTR transcripts in spermatocytes and round sperm cells [19]. The m^6^A recognition protein YTHDC2 can affect the degradation and translation of target genes. YTHDC2 knockout results in the inability of the testis to produce sperm, thereby blocking fertilization [24]. Therefore, yak testis development and spermatogenesis might mainly be regulated by demethylases, and YTHDC2 might affect normal meiosis by regulating mRNA degradation and decay.

We found several m^6^A modification sites in yak testicular tissue by MeRIP-seq. After functional enrichment analysis of DMPs screened before and after sexual maturity, we found that the differential expression of m^6^A-methylated mRNAs involved multiple biological pathways, many of which were closely related to male reproduction. Adjacent Sertoli cells in the testis form a blood–testis barrier (BTB) by tightly connecting the adherens junction and gap junction to prevent the immune system from attacking its own germ cells [68]. Simultaneously, the dynamic changes in BTB provide conditions for the proliferation and differentiation of germ cells [69]. Destruction of testicular immune homeostasis may lead to orchitis, a cause of male infertility [70]. The Wnt, PI3K, and mTOR signaling pathways are involved in many stages of male reproduction and regulate spermatogenesis and testicular endocrine function [71,72]. In particular, mTOR plays an important role in the maintenance and differentiation of spermatogonia and in regulating the redox balance and metabolic activity of Sertoli cells [72].

The transcriptome results showed that DEGs screened after sexual maturity were closely related to calcium ion binding, spermatogenesis, cell differentiation, cilium assembly, and flagellated sperm motility and were significantly enriched in ECM–receptor interaction, cortisol synthesis and secretion, and aldosterone synthesis and secretion pathways. In mammals, cilia are essential for development, sensation, cell signaling, and sperm motility. Calcium-binding proteins regulate signaling pathways necessary for ion metabolism and sperm maturation events. Shawki et al. identified EFCAB2 as a new calcium-binding protein in mouse testis, which may regulate sperm flagellar movement [73]. In our results, the *EFCAB2* gene was significantly up-regulated in the testis of sexually mature yaks. We also found that the *WDR5* gene is one of the up-regulated DEGs. WDR protein plays an important role in spermatocyte division, sperm flagellar assembly, and head formation [74]. ECM includes the seminiferous tubule basement membrane and peritubular intercellular matrix. It contains many cytokines and their receptors. As a component of the seminiferous tubule basement membrane and lamina propria, it participates in BTB formation, is closely related to the composition of the spermatic cord, and is involved in spermatogenesis and spermatogenic cell differentiation [75]. Cortisol excites the development of spermatogonia in an androgen-independent manner and affects the number of sperm in the testis by promoting meiosis and spermatogenesis [76]. Overall, according to the results of this study, it is speculated that DEGs may be involved in biological processes such as mitosis, meiosis, cell differentiation, flagellated sperm motility, and metabolism during sexual maturity of yak testis.

Combined analysis results of m^6^A-seq and RNA-seq showed that m^6^A methylation was specifically expressed in yak testicles at different developmental stages, and these regulatory factors were crucial for maintaining spermatogenesis. BRCA1 is a nuclear phosphoprotein expressed in a broad spectrum of tissues during cell division [77]. *BRCA1* is expressed in the spermatogonia and premeiotic spermatids in the testis [78], and BRCA1-deficient male mice are sterile due to meiotic failure [52]. *CREBBP* is expressed in Sertoli cells and undifferentiated spermatogonia and may participate in regulating the activation of the follicle-stimulating hormone-mediated transferrin promoter in the testis by connecting bHLH and CREB activities [79]. CREBBP can regulate the expression of testicular determinants during gonadal development in the embryonic stage, thus directly affecting testis development [53]. STAT3 is a member of the signal transduction and transcriptional activator protein family, which can be activated by extracellular growth factors and cytokines and participate in the regulation of cell proliferation, differentiation, and apoptosis [80]. In mouse testis, *STAT3* is widely expressed in the supporting cells of the entire seminiferous tubules, and the expression level of *STAT3* in the seminiferous tubules is closely related to spermatogenesis [54]. When STAT3 phosphorylation is inhibited, spermatogenic cell apoptosis increases [81]. SMAD4 protein is a co-transducer of the transforming growth factor (TGF)-β superfamily and participates in the signal transduction of all members of the TGF-β family. *SMAD4* is expressed in mammalian testis and plays an important role in testis development and spermatogenesis [55]. Our results suggest that m^6^A methylation affects testis development and spermatogenesis by regulating the translation and degradation of *BRCA1*, *CREBBP*, *STAT3*, and *SMAD4*.

## 5. Conclusions

In summary, we found that a large number of m^6^A modifications occurred during the sexual maturity of yak testis, and the m^6^A level of yak testicular tissue was significantly higher after sexual maturity than at the juvenile stage. By measuring the expression of methylation-related enzymes, we found that yak testis development and spermatogenesis were mainly regulated by demethylases. The differential expression of m^6^A-methylated mRNAs involved multiple biological pathways, such as Wnt, PI3K, and mTOR signaling pathways, and participated in germ cell proliferation, cell differentiation, spermatogenesis, sperm motility, and other processes. Finally, we also screened the key regulatory factors affecting testis development. Subsequently, the screened key regulatory factors will be verified at the cellular level, and the molecular mechanism of these regulatory factors regulating spermatogenesis through m^6^A will be further explored. This study provides new insights into further understanding the epigenetic molecular regulation mechanism of testicular development.

## Figures and Tables

**Figure 1 animals-13-02815-f001:**
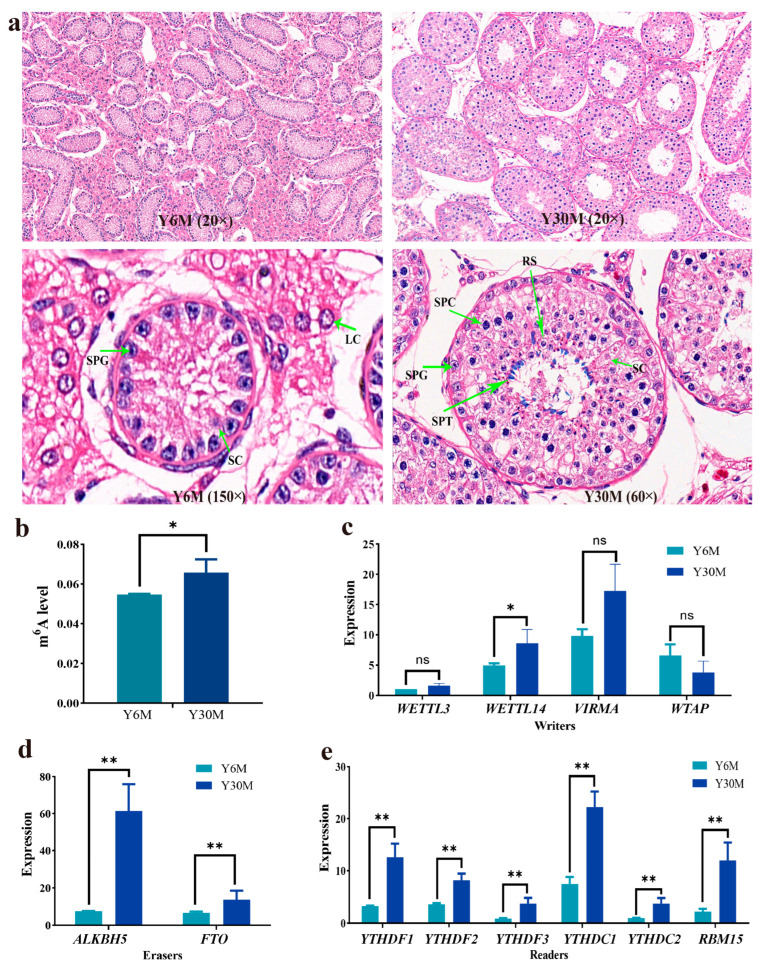
Tissue structure and relative m^6^A level of yak testis at different stages. (**a**) Tissue structure of yak testis at different stages. SPG—spermatogonia; SC—Sertoli cell; LC—Leydig cell; SPC—spermatocyte; RS—round sperm; SPT—spermatid. (**b**) Relative m^6^A levels of yak testis at different stages. (**c**) Expression of writers in yak testis at different stages. (**d**) Expression of erasers in yak testis at different stages. (**e**) Expression of readers in yak testis at different stages (* represents a significant difference in gene expression between the two groups (*p* < 0.05), ** represents a very significant difference in gene expression between the two groups (*p* < 0.01), and ns represents no significant difference in gene expression between the two groups (*p* > 0.05)).

**Figure 2 animals-13-02815-f002:**
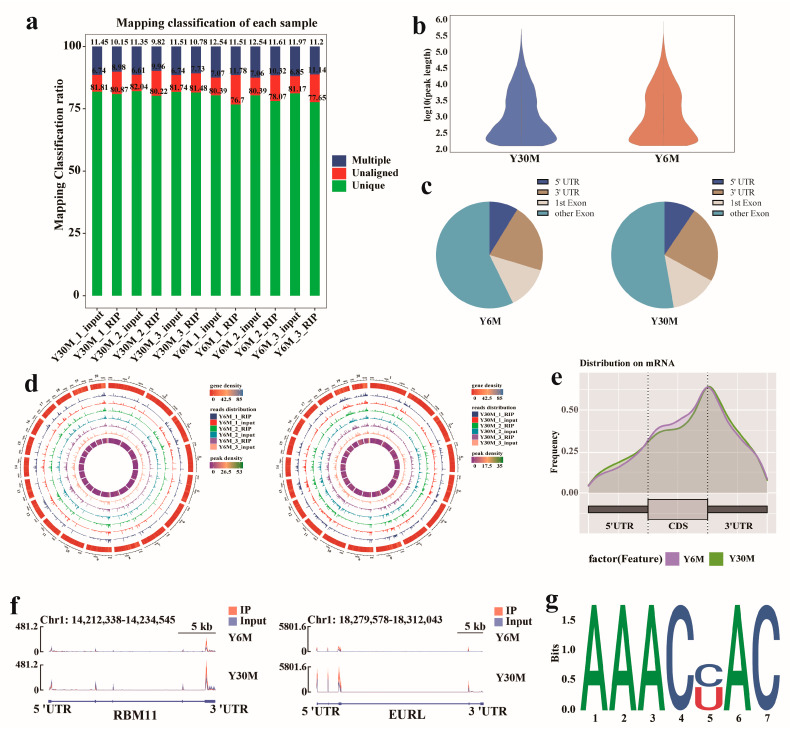
Characteristics of m^6^A in yak testis before and after sexual maturity. (**a**) Distribution of reference genome alignment. (**b**) Violin plot of m^6^A peak length distribution. (**c**) Annotation classification of the peak in the gene functional element region. (**d**) Distribution circle of reads on the reference genome. (**e**) Statistics of the concentration location of the m^6^A peak in yak testis. (**f**) Data visualization analysis of differential m^6^A peaks in the selected mRNAs (RBM11 and EURL). (**g**) Motif sequence enriched by the m^6^A peak.

**Figure 3 animals-13-02815-f003:**
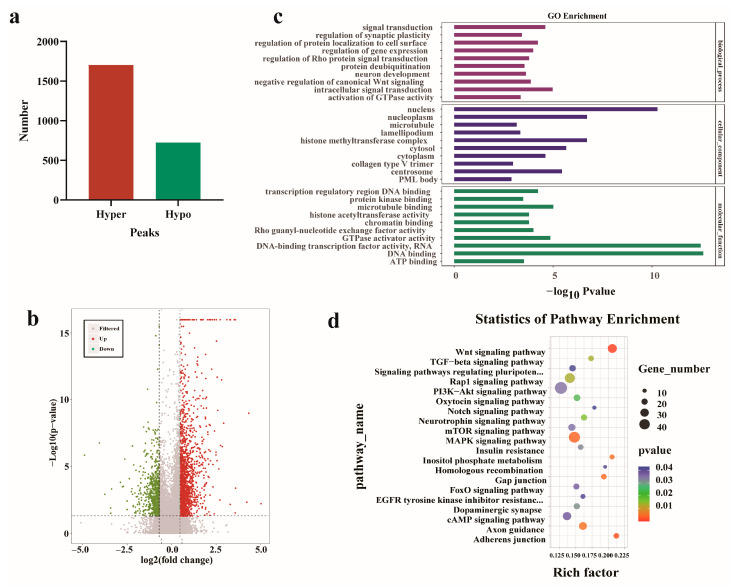
Results of differentially methylated peaks (DMPs) before and after sexual maturity of yak testis. (**a**) Statistical histogram of DMPs. (**b**) Volcano plot of DMP distribution. (**c**) GO enrichment analysis of DMPs. (**d**) KEGG enrichment analysis of DMPs.

**Figure 4 animals-13-02815-f004:**
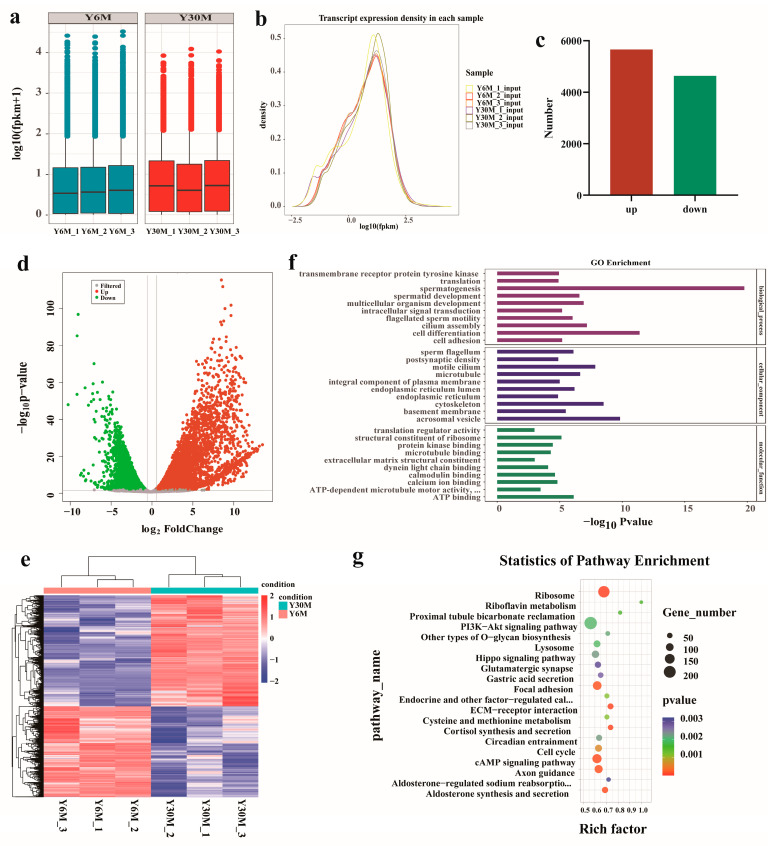
Results of differentially expressed genes (DEGs) before and after sexual maturity of yak testis. (**a**) Box diagram of gene FPKM value of each testicular sample. (**b**) Density distribution curve of gene FPKM value of each testicular sample. (**c**) Statistical histogram of DEGs. (**d**) Volcano plot of DEG distribution. (**e**) Cluster heatmap of DEGs. (**f**) GO enrichment analysis of DEGs. (**g**) KEGG enrichment analysis of DEGs.

**Figure 5 animals-13-02815-f005:**
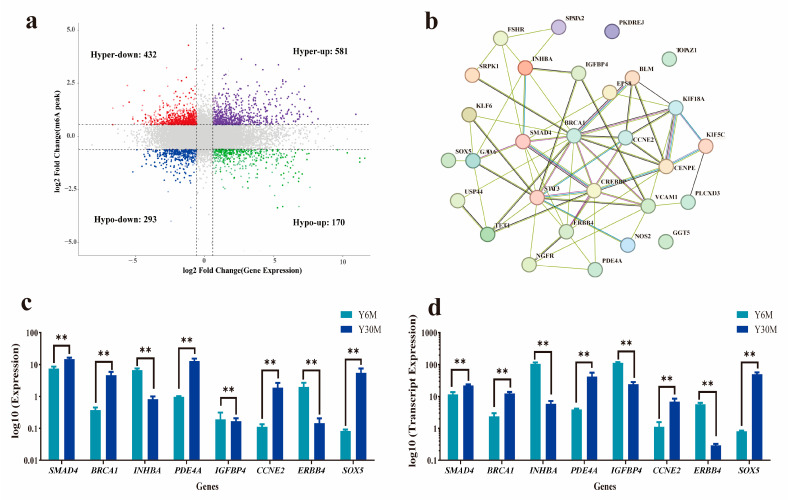
Combined analysis of m^6^A-seq and RNA-seq data. (**a**) Combined four-quadrant map of differentially methylated peaks and differentially expressed genes. (**b**) Network diagram of 30 differentially co-expressed genes. (**c**) qRT-PCR results of DEGs. (**d**) Transcriptome sequencing expression of DEGs (** represents a very significant difference in gene expression between the two groups (*p* < 0.01)).

**Table 1 animals-13-02815-t001:** The 30 genes with significant changes in the transcriptional abundance of m^6^A and mRNA in the yak testis.

Name	Pattern	Chr	Start	End	log_2_FC (m^6^A)	log_2_FC (mRNA)	*p*-Value
*PKDREJ*	hyper-up	5	125,560,273	125,560,773	2.62	5.63	2.65 × 10^−55^
*SOX5*	hyper-up	5	16,470,677	16,470,826	1.08	5.87	1.96 × 10^−47^
*INHBA*	hyper-down	4	44,549,993	44,550,142	0.91	−4.30	1.08 × 10^−35^
*SRPK1*	hypo-up	24	26,610,618	26,610,814	−0.68	3.99	4.83 × 10^−28^
*ERBB4*	hypo-down	2	41,164,674	41,164,923	−0.91	−4.36	1.56 × 10^−27^
*USP44*	hypo-up	5	82,720,213	82,720,562	−1.18	5.95	3.89 × 10^−21^
*KIF5C*	hypo-up	2	98,167,244	98,168,507	−0.69	4.00	7.66 × 10^−21^
*TET1*	hyper-down	27	24,532,195	24,532,594	0.86	−3.90	5.95 × 10^−15^
*NOS2*	hypo-down	19	49,714,496	49,720,399	−0.64	−3.58	1.03 × 10^−14^
*IGFBP4*	hypo-down	19	25,616,251	25,617,161	−0.67	−2.38	2.37 × 10^−14^
*KLF6*	hyper-down	12	36,414,828	36,415,221	1.10	−2.31	8.18× 10^−55^
*PLCXD3*	hypo-down	23	50,340,333	50,417,181	−1.06	−4.24	2.38 × 10^−13^
*NGFR*	hyper-down	19	29,717,922	29,719,581	0.62	−2.57	3.59 × 10^−12^
*GATA6*	hyper-down	21	30,279,347	30,302,478	0.67	−2.60	7.39 × 10^−12^
*TOPAZ1*	hypo-up	22	49,897,438	49,897,685	−1.74	3.79	1.22 × 10^−11^
*FSHR*	hyper-down	9	51,638,767	51,638,966	1.17	−2.66	1.27 × 10^−11^
*PDE4A*	hypo-up	8	15,331,939	15,332,787	−0.63	3.41	1.29 × 10^−11^
*BRCA1*	hyper-up	19	22,819,897	22,820,196	1.03	2.27	1.21 × 10^−9^
*SPATA2*	hypo-up	12	74,095,786	74,096,077	−0.67	2.52	6.29 × 10^−9^
*STAT3*	hypo-down	19	23,617,050	23,617,579	−1.47	−1.80	2.49 × 10^−7^
*VCAM1*	hypo-down	3	28,996,951	28,999,908	−2.03	−2.94	4.33 × 10^−7^
*CCNE2*	hypo-up	18	61,692,482	61,693,870	−0.84	2.50	1.43 × 10^−6^
*GGT5*	hypo-down	16	79,102,897	79,103,146	−1.30	−1.65	4.48 × 10^−6^
*VDAC3*	hypo-up	28	16,482,220	16,482,368	−0.61	1.78	1.97 × 10^−5^
*EPS8*	hypo-down	5	8,127,710	8,130,594	−1.72	−1.40	5.12 × 10^−5^
*CREBBP*	hyper-down	26	44,363,116	44,364,442	0.77	−1.16	0.00016
*BLM*	hyper-up	17	22,597,323	22,597,571	0.72	1.55	0.00033
*SMAD4*	hyper-down	21	48,509,408	48,509,706	0.72	−1.10	0.00047
*KIF18A*	hyper-up	13	9,570,719	9,570,965	0.65	2.25	0.0006
*CENPE*	hyper-up	6	105,142,758	105,146,881	0.82	1.41	0.0345

## Data Availability

The dataset supporting the conclusions of this article is available in the GEO database: GSE229363. The data generated or analyzed during this study are included in this published article and its Appendix A.

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
