# Peer review of "Transcriptome Studies Reveal the N6-Methyladenosine Differences in Testis of Yaks at Juvenile and Sexual Maturity Stages"

_animals, 2023, doi:10.3390/ani13182815_

Round 1
Reviewer 1 Report
I have indicated my comments and suggestions in the attached report to help the authors improve on the quality of the manuscript.

The quality of the English language is good enough and the reader can understand what the authors are communicating in the manuscript.
Author Response
The authors have produced a good manuscript based on their experiment on genomic differences of juvenile and matured yak testis. The authors provide adequate information on the important role of the testis in mammalian reproduction and M6A role in gene expression and regulation. A problem statement, objectives and justification for the study are also clearly indicated. The methodology is clear and easy to follow with generally satisfactory presentation of results.
However, authors are invited to renumber Supplementary Tables to appear in the text chronologically.
Thank you for your suggestion. We have renamed and sorted the supplementary tables in the order they appear in the manuscript.
In the discussion the authors offer good explanations for their findings in line to the known physiology of male reproduction in mammals. Authors need to improve on the conclusions of the study and provide some directions for future research.
Thank you for your suggestion. We have added “Finally, we also screened the key regulatory factors affecting testis development., including BRCA1, CREBBP, STAT3, and SMAD4. Subsequently, the screened key regulatory factors will be verified at the cellular level, and the molecular mechanism of these regulatory factors regulating spermatogenesis through m6A will be further explored. This study provides new insights into further understanding the epigenetic molecular regulation mechanism of testicular development” in the conclusion.
The manuscript is generally well written and my comments in Table 1 is to help the authors to improve on the quality of the manuscript to enable readers derive maximum understanding of the experiment and application of the recommendations made therein.
Table 1. Comments and Suggestions for Authors
14 – 18 Authors should not use unfamiliar terms such as MeRIP-seq, m6A, Wnt, PI3K, mTOR, BRCA1, CREBBP, STAT3 and SMAD4 in the Simple Summary.
Thank you for your comment. We have changed this paragraph to “The methylation level of testicular tissue in young yaks and adult yaks was determined, and the N6-methyladenosine (m6A) level was significantly increased after sexual maturity. Functional enrichment analysis of differentially methylated mRNAs screened between the two groups after sequencing showed that they mainly regulated cell development, spermatogenesis and testicular endocrine function. Functional analysis of the differentially expressed genes screened between the two groups showed that they were involved in the biological processes of mitosis, meiosis, and flagellated sperm motility. Finally, several genes related to spermatogenesis were screened for further verification”.
20 Correct “the key” to read “key”.
Thank you for your comment. We have made modifications according to your suggestion.
36 Not a recommended practice to repeat words used in the title as keywords.
Thank you for your comment. We have changed Keywords to “Spermatogenesis; mRNA; N6-methyladenosine; reproduction”.
39 Edit “animal husbandry resource” to read “animal genetic resource”.
Thank you for your comment. We have made modifications according to your suggestion.
47 Edit “main sexual organ” to read “primary sexual organ”.
Thank you for your comment. We have made modifications according to your suggestion.
55 Correct “the key” to read “key”.
Thank you for your comment. We have made modifications according to your suggestion.
93 Edit “a basis data” to read “preliminary data”.
Thank you for your comment. We have made modifications according to your suggestion.
116-117 Reword the sentence for clarity.
Thank you for your comment. We have changed this sentence to “The steps of RNA extraction and quality control, library construction, sequencing and bioinformatics analysis refer to the methods reported by Guo et al. [29]”.
118 One expects that the first supplementary table that will be referred to in the text will be “Supplementary Table 1” not “Supplementary Table 7”. Unless there is a special reason for doing this, authors should renumber the supplementary tables accordingly.
Thank you for your comment. We have renamed and sorted the supplementary tables in the order they appear in the manuscript.
127 Authors should be more precise regarding which supplementary material they are referring to here.
Thank you for your comment. We have changed “supplementary material” to “supplementary file 1”.
132-133 Provide appropriate reference(s) to support the method cited here.
Thank you for your comment. We have added corresponding references here.
134 Provide year of release of the statistical software indicated here.
Thank you for your comment. We have added the release year 2013 of SAS 9.4 to the main text.
134 Correct level of statistical significance to read (p ≤ 0.05)ï¼›157-159 Correct sentence to read “The m6A level of yak testicular tissue after sexual maturity was significantly (p < 0.05) higher than at the juvenile stage (Figure 1b)ï¼›160-162 Correct sentence to read “The expression of FTO and ALKBH5 was significantly (p < 0.01) higher in mature than in juvenile yak testis (Figure 1d), and the expression of all recognition proteins detected was also significantly (p < 0.01) increased (Figure 1e)ï¼›163-165 Correct sentence to read “However the expression of METTL3, WTAP, and VIRMA showed no significant (p > 0.05) difference in yak testis before and after sexual maturity (Figure 1c).
Thank you for your comment. We have made modifications according to your suggestion.
217 and 219 Supply appropriate references for the methods “Functional enrichment analysis” and “Gene Ontology (GO) analysis”.
Thank you for your comment. We have added corresponding references here.
281-282 Authors should be specific if the “certain degree of correlation” was a significant (p ≤ 0.05) one.
Thank you for your comment. We calculated the correlation between m6A modification and gene expression, r = 0.22, indicating that there is a weak correlation between the two omics.
301-303 Information on current challenges of male yak fertility and possible impacts on agricultural production in China with appropriate references here will help improve on the justification of the study.
Thank you for your comment. We have added “The production and reproduction of male yaks are affected by genetic, environmental and management factors, including season, light, temperature, nutrition and breeding techniques [56]. Due to the long-term domestic domestication, climate and forage resources, yaks only have sexual behavior in the estrus season, which greatly limits the reproductive activity of male yaks. Compared with common cattle, the mating ability of male yak is weaker. In addition, the current yak breeds are seriously degraded, and the efficiency of artificial insemination of yaks in pastoral areas is extremely low, which also poses challenges to yak breeding and production [57]. The low efficiency of yak reproductive performance not only affects the improvement of yak, the breeding of new varieties, but also limits the large-scale promotion of yak, which has a certain impact on the economy of herdsmen, and affects the development of animal husbandry economy in the Qinghai-Tibet Plateau” in line 333-344.
316 “Puberty is late in male yak” – Authors should indicate the age at puberty and support with appropriate reference(s).
Thank you for your comment. We have revised this sentence to “The sexual maturity of male yaks is generally 2-3 years old”.
322-325 Reference(s) needed to support this sentence.
Thank you for your comment. We have added corresponding references here.
374-375 Reference(s) needed to support this sentence.
Thank you for your comment. We have added corresponding references here.
376-377 “…, and BRCA1-deficient male mice are sterile due to meiosis” – explain further or reword for clarity.
Thank you for your comment. We have revised this sentence to “and BRCA1-deficient male mice are sterile due to meiotic failure”.
402-403 The last sentence of the conclusion reads like part of the “methodology”.
394-403 Authors should improve on the Conclusion and provide some practical recommendations regarding future research in this area of study.
Thank you for your comment. We have added “Finally, we also screened the key regulatory factors affecting testis development., including BRCA1, CREBBP, STAT3, and SMAD4. Subsequently, the screened key regulatory factors will be verified at the cellular level, and the molecular mechanism of these regulatory factors regulating spermatogenesis through m6A will be further explored. This study provides new insights into further understanding the epigenetic molecular regulation mechanism of testicular development” in the conclusion.
404 Incomplete statement should be completed.
Thank you for your comment. We have supplemented according to the requirements.
Reviewer 2 Report
The authors studied the level of m6A in yak testis at juvenile and sexual maturity stages. In this paper, the m6A level and the expression level of methylation-related enzymes were determined, and the m6A sequencing of yak testicular tissues in two periods was carried out. It was found that the differential expression of m6A methylation mRNA involved a variety of biological pathways, and the main regulatory factors affecting testicular development were discussed. In general, the author 's manuscript is comprehensive in terms of elaboration. The analysis of the data is well done, but there are only a few minor problems that need to be modified.
1. Why did the experiment select yak testis tissue in these two periods?
2. The last sentence on lines 43-44 is problematic. Please revise it.
3. Lines 146-148,“Spermatogonia and spermatocytes in different division stages increased, and more round sperm cells and spermatids were observed near the lumen”. What does this mean? The statement is not clear.
4. Lines 304,“Puberty is late in male yak”. The expression of this sentence is not appropriate. It cannot be said that yaks are late puberty, and each species has its own developmental history.
5. There are downregulated and upregulated forms in the text, and the two words should be connected with “-”, full-text checked.
6. BRCA1, CREBBP, STAT3, SMAD4, these genes appear for the first time in the full text should be listed in full name. In addition, all genes should be italicized.
7. Lines 134, P values should be italicized.
8. In 3.4. Analysis of differentially methylated peaks (DMPs) of yak testis before and after sexual maturity. I think it would be more appropriate to mention the conditions for identifying DMPs and the functional enrichment analysis in Materials and Methods. Moreover, the conditions for identifying DEGs should also be mentioned in Materials and Methods (Lines 249-250).
9. In Figure 3b, please provide clearer figure.
10. The “* *” symbol in the histograms in Figures 1 and 5 does not indicate its statistical significance, and should be added to the diagram notes.
All genes and P values should be italicized.
Author Response
The authors studied the level of m6A in yak testis at juvenile and sexual maturity stages. In this paper, the m6A level and the expression level of methylation-related enzymes were determined, and the m6A sequencing of yak testicular tissues in two periods was carried out. It was found that the differential expression of m6A methylation mRNA involved a variety of biological pathways, and the main regulatory factors affecting testicular development were discussed. In general, the author 's manuscript is comprehensive in terms of elaboration. The analysis of the data is well done, but there are only a few minor problems that need to be modified.
- Why did the experiment select yak testis tissue in these two periods?
Thank you for your question. The purpose of this experiment is to investigate the differences of m6A methylation in the testicular tissue of yaks before and after sexual maturation. Therefore, the experiment selectively selected yaks before sexual maturity ( 6 months old ) and sexual maturity ( 30 months old ).
- The last sentence on lines 43-44 is problematic. Please revise it.
Thank you for your comment. We have revised this sentence to “Therefore, the study of yak reproductive performance is helpful to improve the economic development and people 's quality of life in the Qinghai-Tibet Plateau”.
- Lines 146-148, “Spermatogonia and spermatocytes in different division stages increased, and more round sperm cells and spermatids were observed near the lumen”. What does this mean? The statement is not clear.
Thank you for your comment. We have revised this sentence to “After sexual maturity, the number of spermatogonia and spermatocytes in the testicles of yaks increases, and more round sperm cells and spermatids were observed near the lumen”.
- Lines 304, “Puberty is late in male yak”. The expression of this sentence is not appropriate. It cannot be said that yaks are late puberty, and each species has its own developmental history.
Thank you for your comment. We have revised this sentence to “The sexual maturity of male yaks is generally 2-3 years old”.
- There are downregulated and upregulated forms in the text, and the two words should be connected with “-”, full-text checked.
Thank you for your comment. We have made modifications in the full text.
- BRCA1, CREBBP, STAT3, SMAD4, these genes appear for the first time in the full text should be listed in full name. In addition, all genes should be italicized.
Thank you for your comment. We have added the full names of these genes when they first appeared in lines 313-316.
- Lines 134, P values should be italicized.
Thank you for your comment. We have modified it to italicized.
- In 3.4. Analysis of differentially methylated peaks (DMPs) of yak testis before and after sexual maturity. I think it would be more appropriate to mention the conditions for identifying DMPs and the functional enrichment analysis in Materials and Methods. Moreover, the conditions for identifying DEGs should also be mentioned in Materials and Methods (Lines 249-250).
Thank you for your comment. We added “Differential peak detection analysis was performed using MeTDiff software (P≤0.05; |FC|≥1.5). In the transcriptome sequencing analysis, the fragments per kilobase of transcript per million mapped reads (FPKM) value of the gene was used to represent the expression level of the gene. The screening condition for differentially expressed genes between samples was (P≤0.05; |FC|≥1.5). The condition for screening significantly enriched items in Gene Ontology ( GO ) and Kyoto Encyclopedia of Genes and Genomes ( KEGG ) analysis was P < 0.05” in line 139-145 in Materials and Methods.
- In Figure 3b, please provide clearer figure.
Thank you for your comment. We have replaced Figure 3b with a clear image.
- The “* *” symbol in the histograms in Figures 1 and 5 does not indicate its statistical significance, and should be added to the diagram notes.
Thank you for your comment. We have added “* represents a significant difference in gene expression between the two groups (P < 0.05), ** rep-resents a very significant difference in gene expression between the two groups (P < 0.01), ns rep-resents no significant difference in gene expression between the two groups (P > 0.05).” to the notes in Figure 1 and Figure 5.
Reviewer 3 Report
The topic considered in the manuscript is relevant and promising.
The study of the processes of methylation - demethylation as regulators of gene expression is very significant, since science is just beginning to learn these mechanisms.
However, there is a very serious error in the research methodology. This is the time for taking biological material. Yaks are animals with a pronounced seasonality of breeding. In the same animals, the processes of spermatogenesis are significantly related to the season in which the rut occurs. Only in these one and a half - two months the process of spermatogenesis occurs in the testes, in the rest of the year spermatogenesis completely fades. And it stops so much that when trying to get sperm from the epididymis (post mortem), it is possible to detect single (no more than 10) dead spermatozoa. That is, at the time of the absence of the rut, the genes responsible for spermatogenesis are not expressed. And this happens due to the methylation of these genes. It is not clear at what point the authors of the manuscript took biomaterial from adult yaks. If we take into account the fact that males during the rut are extremely aggressive and it is impossible to take biological material from them, only if the postmortem, then we can conclude that the material was taken from males during the rest period of the reproductive system. Thus, it can be assumed that the processes of methylation and a decrease in the expression level of spermatogenesis genes are associated not with sexual maturity, as the authors suggest, but with the period of seasonal cessation of sexual activity.
The authors should conduct these studies on three groups of animals: immature yaks, sexually mature yaks in the rut, sexually mature yaks in the stage of sexual dormancy. A comparative analysis of the expression and methylation of genes of reproductive function in these three groups will provide a complete picture of the regulation of reproductive processes in animals with a pronounced seasonality of reproduction.
Author Response
The topic considered in the manuscript is relevant and promising.
The study of the processes of methylation - demethylation as regulators of gene expression is very significant, since science is just beginning to learn these mechanisms.
However, there is a very serious error in the research methodology. This is the time for taking biological material. Yaks are animals with a pronounced seasonality of breeding. In the same animals, the processes of spermatogenesis are significantly related to the season in which the rut occurs. Only in these one and a half - two months the process of spermatogenesis occurs in the testes, in the rest of the year spermatogenesis completely fades. And it stops so much that when trying to get sperm from the epididymis (post mortem), it is possible to detect single (no more than 10) dead spermatozoa. That is, at the time of the absence of the rut, the genes responsible for spermatogenesis are not expressed. And this happens due to the methylation of these genes. It is not clear at what point the authors of the manuscript took biomaterial from adult yaks. If we take into account the fact that males during the rut are extremely aggressive and it is impossible to take biological material from them, only if the postmortem, then we can conclude that the material was taken from males during the rest period of the reproductive system. Thus, it can be assumed that the processes of methylation and a decrease in the expression level of spermatogenesis genes are associated not with sexual maturity, as the authors suggest, but with the period of seasonal cessation of sexual activity.
The authors should conduct these studies on three groups of animals: immature yaks, sexually mature yaks in the rut, sexually mature yaks in the stage of sexual dormancy. A comparative analysis of the expression and methylation of genes of reproductive function in these three groups will provide a complete picture of the regulation of reproductive processes in animals with a pronounced seasonality of reproduction.
Thank you for your comments. The purpose of this experiment is to explore the difference of M6 A methylation in testicular tissue of yak before and after sexual maturity. Therefore, yaks before sexual maturity (6 months old) and sexual maturity (30 months old) were selected. At present, yak samples are collected from sexually mature yaks in August. At this time, yaks are in the estrus stage. At the same time, through slice observation, 30-month-old yak testis samples can observe spermatogenic cells and sperm in each stage, so the experimental samples meet the experimental requirements. We have added testicular collection time in materials and methods.
Round 2
Reviewer 3 Report
Yes, the manuscript is improved enough to warrant publication in Animals. However, we would like to recommend the authors to pay more attention to the seasonality of yak breeding, the features of spermatogenesis in animals with a pronounced seasonality of reproduction, and to add this information to the manuscript.
Author Response
Thank you for your comment. We have added "Silver fox is a seasonal estrous animal. Cheng et al. [56 ] reported that the testis mass, volume and testosterone content of silver fox increased gradually from septem-ber to january of the next year. The composition and number of spermatogenic cells in the cross section of a single seminiferous tubule on the section gradually increased with time and showed seasonal changes [56 ]. The yak is also a seasonal estrous animal, usually experience estrus during summer." in line 330-335.